

# Measuring walking impairment in patients with intermittent claudication: psychometric properties of the Walking Estimated-Limitation Calculated by History (WELCH) questionnaire

Farhad Rezvani, Martin Härter and Jörg Dirmaier

Department of Medical Psychology, University Medical Center Hamburg-Eppendorf, Hamburg, Germany

## ABSTRACT

**Objectives**. Patient-reported outcome measures can facilitate the assessment of walking impairment in peripheral artery disease patients with intermittent claudication in clinical trials and practice. The aim of this study was to test the psychometric properties of the German version of the 'Walking Estimated-Limitation Calculated by History' (WELCH) questionnaire.

**Methods**. The assessed properties included feasibility, test-retest reliability, construct validity (i.e., convergent, divergent and known-groups validity) and responsiveness using classic psychometric methods. Psychometric properties were tested as part of a randomized controlled home-based exercise trial for patients with symptomatic peripheral artery disease at Fontaine stage IIA/B.

**Results**. Analyses were conducted in subgroups of 1,696 patients at baseline and 1,233 patients at 12-month follow-up (i.e., post-intervention) who completed the WELCH along with a battery of other self-report measures. The WELCH did not exhibit relevant floor or ceiling effects ($< 15\%$ achieved lowest or highest possible scores), showed evidence for good test-retest reliability (ICC $= .81$, 95% CI [.71–.88]) and was found to be well suited for self-completion by patients ($< 5\%$ missing data per item). WELCH scores showed moderate to strong correlations with related measures of walking impairment at both time points (Walking Impairment Questionnaire: $r = .56 - .74$; VascuQoL-25 activity subscale: $r = .61 - .66$) and distinguished well among patients with poor and high quality of life when adjusting for confounders ($t = 13.67$, $p < .001$, $d = .96$). Adequate divergent validity was indicated by a weaker correlation between the WELCH and general anxiety at both time points (GAD-7: $r = -.14\ to\ -.22$). The WELCH improved by 6.61 points (SD $= 17.04$, 95% CI [5.13–8.10], $d = 0.39$) in response to exercise treatment and was able to identify large clinically important improvements observed on the walking distance (AUC $= .78$, 95% CI [.71–.84]) and speed subscales (AUC $= .77$, 95% CI [.68–.86]) of the Walking Impairment Questionnaire.

**Conclusions**. The WELCH is considered a feasible, reliable and valid patient-reported outcome measure for the measurement of walking impairment in patients with peripheral artery disease. The WELCH showed evidence for responsiveness to changes

Corresponding author
Farhad Rezvani, f.rezvani@uke.de

![PeerJ]

in walking impairment, yet further studies are warranted to conclusively determine the WELCH's ability to detect intervention effects.

## INTRODUCTION

Peripheral Artery Disease (PAD) is a global public health problem affecting an estimated 200 million people worldwide and has become one of the leading causes of disability and death over recent decades (*Criqui & Aboyans, 2015*; *Sampson et al., 2014*; *Song et al., 2019*). The most common symptom is intermittent claudication (IC), which refers to cramping leg pain that is caused by exercise due to insufficient blood flow (*Criqui & Aboyans, 2015*; *Aboyans et al., 2018*).

Rapid clinical assessment of walking impairment in symptomatic PAD patients is crucial in vascular surgery practice, providing a clinically relevant endpoint from the patient perspective by reflecting walking difficulties in everyday life and therefore considered a better tool for measuring patient-reported walking ability than functional surrogate endpoints (*Frans et al., 2013*). The 'Walking Estimated-Limitation Calculated by History' (WELCH) questionnaire is a brief patient-reported outcome measure (PROM) instrument that requires minimal completion time for assessing walking capacity, with the intention to be used routinely in clinical practice (*Ouedraogo et al., 2013*). The WELCH has been translated and cross-validated into various languages (*Ouedraogo et al., 2013*; *Abraham et al., 2014a*; *Cucato et al., 2016*; *Abraham et al., 2014b*; *Tew et al., 2014*), has shown good feasibility results as it is easy to score and, compared to other PROMs, considered less prone to errors when self-administered by the patient (*Ouedraogo et al., 2013*; *Ouedraogo et al., 2011*), while correlating well with treadmill walking (*Ouedraogo et al., 2013*; *Tew et al., 2014*; *Henni et al., 2019*; *Fouasson-Chailloux et al., 2015*). To be proposed as a routine tool in the future, however, further external validation in larger samples and other languages are required. The purpose of the current study is, therefore, to psychometrically validate the WELCH in a German cohort of symptomatic PAD patients.

## METHODS

### Design

The WELCH was validated as part of a prospective, randomized controlled trial (RCT) evaluating the effectiveness of a 12-month long home-based exercise program for patients with IC, PAD-TeGeCoach. The study protocol was registered and published elsewhere (ClinicalTrials.gov trial registration: NCT03496948) (*Rezvani et al., 2020*). The study was conducted in accordance with the Declaration of Helsinki and was approved by the ethics committee of the Medical Association of Hamburg (reference number: PV5708). All patients provided written informed consent.
## Study population

Approximately 63,000 PAD patients with IC symptoms aged 35–85 with a clinically confirmed ICD-diagnosis of PAD at Fontaine stadium IIa (*i.e.,* IC > 200 m) or IIb (*i.e.,* IC < 200 m) within the last 36 months were identified using routinely collected health insurance data from inpatient and outpatient encounters. Patients were excluded, if they had asymptomatic PAD within the last 12 months (Fontaine stadium I) or rest pain within the last 36 months (Fontaine stadium III or IV). As the diagnosis of PAD is often flawed, especially in outpatient settings, participants were interviewed about their IC symptoms prior to enrollment to verify the diagnosis of symptomatic PAD. A sample of 1,982 PAD patients (recruitment rate approx. 3.2%) were enrolled and randomized either into the exercise intervention (PAD-TeGeCoach) or the routine care group (see Fig. 1 for RCT flow chart). 11 participants (TeGeCoach $n = 10$; routine care $n = 1$) were withdrawn prematurely after randomization (data deletion request $n = 1$, randomized without informed consent $n = 1$, met exclusion criteria $n = 8$, lack of verification of PAD diagnosis $n = 1$), leading to a final sample size of 1,971 PAD patients (TeGeCoach $n = 984$; routine care $n = 987$).

## Measures

Only RCT data from measures relevant to the present study were used and are described in detail elsewhere (*Rezvani et al., 2020*). While the WELCH was specified in the trial registry as a secondary outcome, the authors failed to include it the study protocol. Notwithstanding this, the (internal and external) validity of the current study is not expected to be compromised as the trial was registered prospectively (*i.e.,* before recruitment).

Participants received a battery of paper-based questionnaires by mail at each time point and were asked to return them using a prepaid envelope. To maximize return rates, participants who had not returned the questionnaire in time received a postal reminder after 2–4 weeks. All participants were followed up at 12 months, irrespective of whether questionnaires had been returned at baseline. The participants could call the study team when they encountered problems completing the questionnaires.

### *'Walking estimated-limitation calculated by history' (WELCH) questionnaire*

The German version used in this study was requested and made available by the authors of the WELCH, which officially has not yet been psychometrically validated (see Supplementary Files). The WELCH was forward translated into German by a native-speaking health professional who was not a member of the WELCH development team, and was then closely back translated into French by the authors to ensure appropriate wording. After two rounds of forward and backwards translation, comparing the original and back-translated French versions, a version was reached that was considered acceptable by the authors. The WELCH consists of four items; items 1–3 are eight-point ordinal items, ranging from "impossible" to "3 h or more", and assess the maximum duration that patients can maintain walking at different speeds in comparison to friends and relatives (*i.e.,* slower/same/faster). Item 4 is a five-point ordinal item, ranging from "much slower" to "faster", and assesses the usual walking *speed* compared to friends and relatives. The

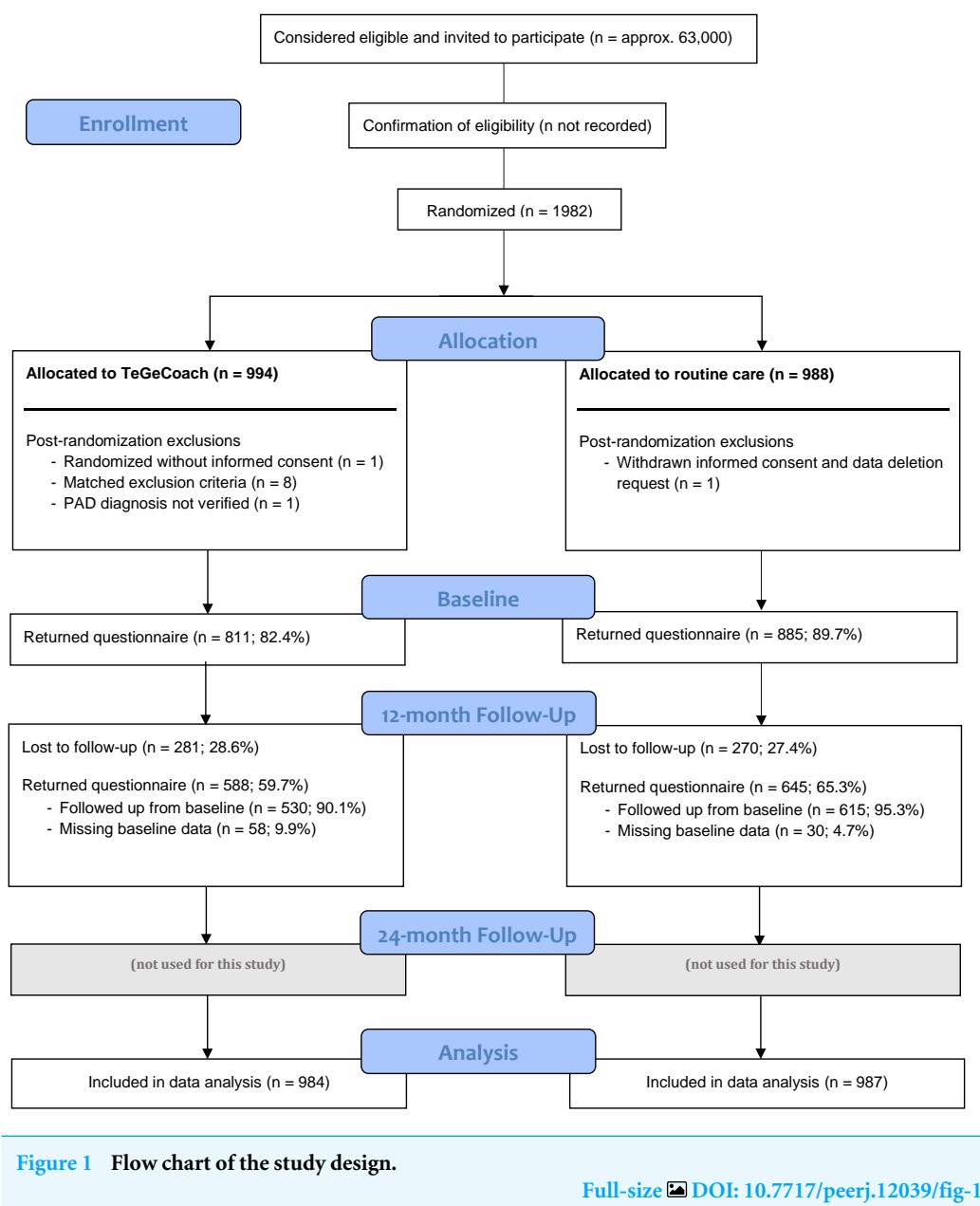

**Figure 1  Flow chart of the study design.**

WELCH score is generated by computing the sum of items 1–3, minus one, and multiplying it by the answer of item 4, *i.e.,* [(Item 1 + Item 2 + Item 3) −1] × Item 4. It is assumed that patients are able to walk at least 30 s at low speed so that the sum of the first three items is never 0. WELCH scores thus range from 0 (*i.e.,* patient is able to walk for a maximum of 30 s at slow speed) to 100 (*i.e.,* patient is able to walk 3 h or more at fast speed). Missing values were handled as indicated by the WELCH authors; for items 1–3 (maximum walking duration), missing values were replaced by the mean of the other two available items (*i.e.,* mean imputation), whereas for item 4, missing values were automatically replaced

by 3 (*Ouedraogo et al., 2013*). The German version of the WELCH can be found in the Supplementary Files.

### Walking Impairment Questionnaire (WIQ)

The Walking Impairment Questionnaire (WIQ) is considered a reliable and valid instrument for assessing walking impairment for different degrees of difficulty across three domains: walking distance, walking speed and stair-climbing (*Regensteiner et al., 1990*; *McDermott et al., 1998*; *Sagar et al., 2012*; *Tew et al., 2013*). WIQ scores are considered responsive to the effect of treatment (*Regensteiner et al., 1990*; *Nicolai et al., 2009*) and are strongly correlated with maximum walking distance (*Frans et al., 2013*; *Tew et al., 2013*), objective measures of walking impairment (*McDermott et al., 1998*), as well as the ankle-brachial index (*Myers et al., 2008*). The proportion of missing values was < 5% for all WIQ items. Item and scale-level missing data were not imputed.

### Kings College Vascular Quality of Life questionnaire (VascuQoL-25)

The VascuQoL-25 evaluates PAD-specific quality of life (QOL) and is divided into five subscales: pain, symptoms, activities, social, and emotional. Although designed to measure health-related QOL in PAD patients, all subscale scores and the composite score strongly correlate with functional status outcomes (*Morgan et al., 2001*; *Mehta et al., 2006*). In addition, the activity subscale was suggested to reflect physical functioning in patients with IC according to Wilson and Cleary's model for health-related QOL (*Wilson & Cleary, 1995*; *Conijn et al., 2015*), and was therefore considered as a valid comparator instrument. The VascuQoL-25 has also been shown to be responsive to changes in disease severity in PAD patients (*Mehta et al., 2006*; *de Vries et al., 2005*). The proportion of missing values was < 5% for all VascuQoL-25 items and were mean-imputed.

### General Anxiety Disorder Scale (GAD-7)

The GAD-7 was shown to be a reliable and valid self-administered instrument with seven items for screening general anxiety by measuring symptom severity in the last two weeks (*Spitzer et al., 2006*; *Lowe et al., 2008*). Items are scored from zero to three on a 4-point scale from 'not at all' to 'nearly every day'. A study conducted in the German general population has found good psychometric properties for the GAD-7 (*Lowe et al., 2008*). Item-level missing data were not imputed.

## Statistical analysis

The psychometric properties of the WELCH were assessed in accordance with the standards set out by the COSMIN group, which guide the development of rigorous methods to investigate psychometric properties of PROMs (*Mokkink et al., 2010*; *Mokkink et al., 2019*). When available, measurement properties were tested against criteria proposed for good measurement properties of health status questionnaires (*Terwee et al., 2007*), which were also used in a broad systematic review examining the psychometric validity of PROMs for measuring IC (*Conijn et al., 2015*).

*Floor and ceiling effects* were assessed by examining frequency distributions of each item and were considered to be present if > 15% of the study sample achieved the lowest or

highest possible score, respectively (*Terwee et al., 2007*). Feasibility of the WELCH was assessed based on the number of missing values in the study sample and per item before imputation. As a rule of thumb, multiple imputation is not considered necessary and single imputation deemed sufficient, if the proportions of missing data are $< 5\%$ (*Schafer, 1999*); therefore, the acceptable proportion of missing data for each item was set at $< 5\%$.

The *test-retest* reliability of the WELCH was assessed based on a group of 67 PAD patients at 12-month follow-up who were instructed to fill out the questionnaire again after two weeks under the same condition (*i.e.*, self-completed at home and returned *via* mail), with patients assumed to be stable during this time period without remembering their exact responses from two weeks earlier. To determine the level of consistency between these two time points (*i.e.*, stability of repeated measurements), the intraclass correlation coefficient (ICC) and the 95% confidence interval (CI) were calculated using a two-way mixed effects model for single measures with absolute reliability. Test-rest reliability of the WELCH was established when the ICC is $> .70$ (*Conijn et al., 2015*; *Terwee et al., 2007*).

Construct validity of the WELCH was verified with *convergent validity*. Pearson correlation coefficients, with bootstrapped CIs based on $n = 1,000$ samples, were determined between the WELCH and the comparator instruments at baseline and 12-month follow-up (*i.e.*, WIQ, VascuQoL-25 activity subscale), and were deemed satisfactory if there was a strong positive correlation $\geq .50$ (*Conijn et al., 2015*). *Divergent validity* was assessed by testing the association of the WELCH with anxiety, a construct known to be unrelated to symptomatic PAD (*Smolderen et al., 2009*); and was considered satisfactory if the correlation between the WELCH and the GAD-7 was weaker than with the comparator instruments at baseline and 12-month follow-up (*Conijn et al., 2015*). *Known-groups validity* was determined using a *t*-test with degrees of freedom adjusted for unequal variances to examine the extent to which the WELCH can significantly discriminate between PAD patients with poor to moderate (VascuQoL-25 score $\leq 4$) and high QOL at baseline and 12-month follow-up, given that QOL and walking impairment are known to be associated in patients with IC (*Golledge et al., 2020*). The discriminatory ability of the WELCH was also assessed using a multiple linear regression to control for potential confounders between QOL and WELCH scores. The model was controlled for all significant variables reported in Table 1. The sample was assumed to be highly representative of the population of IC patients, making the introduction of collider bias unlikely. In the absence of defined quality criteria for health status questionnaires, known-groups validity was also assessed using Cohen's d effect sizes (small: $d = 0.2$, medium: $d = 0.5$, large: $d = 0.8$).

*Responsiveness* of the WELCH was examined by testing the ability to distinguish patients who have and have not changed after receiving the home-based exercise intervention (TeGeCoach), using the area under the receiver operating characteristics (ROC) curve (AUC) at various threshold settings for minimal clinically important differences (MCIDs) on the WIQ subscales (*Gardner, Montgomery & Wang, 2018*). MCIDs reflect the health status change that patients consider beneficial (*Jaeschke, Singer & Guyatt, 1989*). For the WIQ subscales, these were previously determined based on an anchor-based method assessing physical function quality of life following a 3-month (home or supervised) exercise intervention, with exercise protocols closely resembling the PAD-TeGeCoach
**Table 1** Characteristics of subgroups used in the analyses (total sample at baseline, TeGeCoach group followed through to 12 months, TeGe-Coach and routine care group at 12-month follow-up).

| | Baseline | Followed through to 12 months | 12-month follow-up[*] | |
| --- | --- | --- | --- | --- |
| | Total ($n = 1,971$) | TeGeCoach | TeGeCoach | Routine care |
| *N. of questionnaires received* | 1,696 | 530 | 588 | 645 |
| *Sex*[a] | | | | |
| *Female* | 529 (31.2) | 148 (27.9) | 160 (27.2) | 199 (30.9) |
| *Male* | 1146 (67.6) | 371 (70.0) | 417 (70.9) | 441 (68.4) |
| *No information provided* | 21 (1.2) | 11 (2.1) | 11 (1.9) | 5 (0.8) |
| *Age (in years)*[b] | 66.3 (8.6) | 67.0 (8.2) | 67.1 (8.3) | 67.3 (8.4) |
| *Minimum–Maximum* | 35–81 | 35–81 | 35–81 | 38–81 |
| *BMI*[b] | 28.1 (5.0) | 27.9 (5.4) | 27.8 (5.3) | 28.0 (4.7) |
| *Minimum-Maximum* | 15.0–75.8 | 15.0–75.8 | 14.5–75.8 | 16.8–44.5 |
| *Education*[b] *(multiple choices possible)* | | | | |
| *Apprenticeship* | 1166 (68.8) | 365 (68.9) | 365 (62.1) | 431 (66.8) |
| *College* | 562 (33.1) | 177 (33.4) | 177 (30.1) | 224 (34.7) |
| *University* | 289 (17.0) | 105 (19.8) | 105 (17.9) | 104 (16.1) |
| *Other* | 126 (7.4) | 38 (7.2) | 38 (6.5) | 29 (4.5) |
| *No education* | 67 (4.0) | 22 (4.2) | 22 (3.7) | 14 (2.2) |
| *Income*[b] | | | | |
| *< 500 €* | 34 (2.0) | 5 (0.9) | 5 (0.9) | 12 (1.9) |
| *500€ to 1000€* | 135 (8.0) | 33 (6.2) | 33 (5.6) | 41 (6.4) |
| *1001€ to 1500€* | 215 (12.7) | 58 (10.9) | 58 (9.9) | 69 (10.7) |
| *1501€ to 2000€* | 282 (16.6) | 85 (16.0) | 85 (14.5) | 114 (17.7) |
| *2001€ to 2500€* | 306 (18.0) | 108 (20.4) | 108 (18.4) | 103 (16.0) |
| *2501€ to 3000€* | 242 (14.3) | 79 (14.9) | 79 (13.4) | 95 (14.7) |
| *3001€ to 3500€* | 147 (8.7) | 42 (7.9) | 42 (7.1) | 67 (10.4) |
| *3501€ and more* | 211 (12.4) | 73 (13.8) | 73 (12.4) | 78 (12.1) |
| *No information provided* | 124 (7.3) | 47 (8.9) | 105 (17.9) | 66 (10.2) |
| *Marital status*[b] | | | | |
| *Single* | 113 (6.7) | 29 (5.5) | 29 (4.9) | 36 (5.6) |
| *Married* | 1,088 (64.2) | 366 (69.1) | 366 (62.4) | 412 (63.9) |
| *Divorced/separated* | 297 (17.5) | 77 (14.5) | 77 (13.1) | 99 (15.3) |
| *Widowed* | 167 (9.8) | 46 (8.7) | 46 (7.8) | 61 (9.5) |
| *No information provided* | 31 (1.8) | 12 (2.3) | 70 (11.9) | 37 (5.7) |
| *Number of children*[a*] | 1.7 (1.1) | 1.8 (1.1) | 1.8 (1.1) | 1.7 (1.2) |
| *Minimum–Maximum* | 0–11 | 0–6 | 0–6 | 0–11 |

**Table 1** (*continued*)

| | Baseline | Followed through to 12 months | 12-month follow-up[*] | |
|---|---|---|---|---|
| | Total (*n* = 1,971) | TeGeCoach | TeGeCoach | Routine care |
| ***Profession***[b] *(multiple choices possible)* | | | | |
| *Employed* | 462 (27.2) | 143 (27.0) | 143 (24.3) | 141 (21.9) |
| *Unemployed* | 77 (4.5) | 15 (2.8) | 15 (2.6) | 21 (3.3) |
| *Housewife/househusband* | 61 (3.6) | 14 (2.6) | 14 (2.4) | 32 (5.0) |
| *Retired* | 1,057 (62.3) | 351 (66.2) | 351 (59.7) | 410 (63.6) |
| *Retired early* | 52 (3.1) | 23 (4.3) | 23 (3.9) | 12 (1.9) |
| *Permanently incapacitated for work* | 45 (2.7) | 8 (1.5) | 8 (1.4) | 14 (2.2) |
| ***Diseases***[a] *(multiple choices possible)* | | | | |
| *Myocardial infarction* | 217 (12.8) | 79 (14.9) | 85 (14.5) | 82 (12.7) |
| *Stroke* | 149 (8.8) | 51 (9.6) | 53 (9.0) | 53 (8.2) |
| *Metabolism disorder* | 965 (56.9) | 314 (59.2) | 344 (58.5) | 383 (59.4) |
| *Angina pectoris* | 224 (13.2) | 66 (12.5) | 71 (12.1) | 90 (14.0) |
| *Lung disease* | 271 (16.0) | 71 (13.4) | 79 (13.4) | 117 (18.1) |
| *Heart Failure* | 259 (15.3) | 80 (15.1) | 87 (14.8) | 101 (15.7) |
| *Hypertension* | 1,225 (72.2) | 371 (70.0) | 409 (69.6) | 482 (74.7) |
| *Diabetes* | 437 (25.8) | 139 (26.2) | 151 (25.7) | 182 (28.2) |
| *Cancer* | 155 (9.1) | 51 (9.6) | 55 (9.4) | 64 (9.9) |
| ***Drugs***[a] *(multiple choices possible)* | | | | |
| *Antihypertensive agents* | 1,253 (73.9) | 397 (74.9) | 440 (74.8) | 489 (75.8) |
| *Platelet function inhibitor* | 1,370 (80.8) | 444 (83.8) | 491 (83.5) | 540 (83.7) |
| *Statins* | 983 (58.0) | 322 (60.8) | 356 (60.5) | 397 (61.6) |
| ***Revascularization***[a] | | | | |
| *Yes* | 499 (29.4) | 182 (34.3) | 192 (32.7) | 185 (28.7) |
| *No* | 934 (55.1) | 268 (50.6) | 310 (52.7) | 364 (56.4) |
| *No information provided* | 263 (15.5) | 80 (15.1) | 86 (14.6)) | 96 (14.9) |
| ***Group heart rate training***[a] | | | | |
| *Yes* | 221 (13.0) | 79 (14.9) | 85 (14.5) | 90 (14.0) |
| *No* | 1,438 (84.8) | 440 (83.0) | 492 (83.7) | 538 (83.4) |
| *No Information provided* | 37 (2.2) | 11 (2.1) | 11 (1.9) | 17 (2.6) |
| ***Nationality***[a] | | | | |
| *German* | 1,596 (94.1) | 498 (94.0) | 498 (84.7) | 585 (90.7) |
| *Other* | 41 (2.4) | 11 (2.1) | 11 (1.9) | 11 (1.7) |
| *No information provided* | 59 (3.5) | 21 (4.0) | 79 (13.4) | 49 (7.6) |

**Notes.**

[a]Categorical variables: *n* (%).

[b]Quantitative variables: M (SD).

[*]Information on *education, income, marital status, number of children, nationality* and *occupation* was collected at baseline only, and was not available from the 88 participants who completed only the 12-month follow-up questionnaire. These patients were included in *no information provided*.

intervention used in this study (*i.e.,* intermittent walking to near maximal claudication pain while using an activity monitor during exercise sessions) (*Gardner et al., 2014*; *Gardner et al., 2011*). An AUC of ≥ .70 was considered to indicate adequate responsiveness (*Conijn et al., 2015*; *Terwee et al., 2007*). In addition, standardized effect sizes (d, baseline –12-month follow-up) were calculated for the WELCH and the convergent measures.

Analyses were performed using all available data at baseline and during 12-month follow-up. Statistical analyses were performed using SPSS version 25 (IBM Corporation, Armonk, New York, United States). Values of $p < .05$ (two-sided) were considered statistically significant.

## RESULTS

Self-reported sociodemographic and clinical information of the PAD patients were collected at both time points and are presented in Table 1, grouped by time point and subgroups used in the analyses to allow tracking the pattern of missing data (see supplementary materials for an extended version of Table 1). Of those enrolled ($N = 1,971$), 1,696 patients returned their questionnaires at baseline (response rate: 86%). 551 patients were lost to 12-month follow-up, while 1,145 were followed up through 12-month follow-up. 88 patients returned their questionnaire only at 12-month follow-up, resulting in a sample size of 1,233 patients at 12-month follow-up (response rate: 63%). The response rates fall within the usual range of mail surveys (*Asch, Jedrziewski & Christakis, 1997*). Reasons for attrition were not identified, but may be attributed to the patient's right to withdraw at any time without having to give a reason and without penalty. The sample sizes, per analysis, using the RCT data are considered excellent for evaluating the psychometric properties of the WELCH (*Frost et al., 2007*).

### Feasibility, and floor and ceiling effects

Score distributions and missing values per item before imputation are presented in Table 2. At baseline, the total amount of missing data was < 5% per item. A total of 1,611 (95.0%) filled out the WELCH completely at baseline, 79 patients (4.7%) filled it out partially, while only 6 WELCH questionnaires were returned completely empty (0.4%). The number of missing values at 12-month follow-up was similarly low irrespective of study group (Table 2), indicating excellent feasibility. In addition, the number of patients scoring the lowest or highest possible score was < 15% in all items, indicating that there were no floor or ceiling effects irrespective of study group.

### Test-retest reliability

A group of 67 PAD patients filled out the WELCH twice within two weeks at 12-month follow-up. The ICC for the WELCH score was .81 (95% CI [.71–.88]), which indicates good test–retest reliability.

### Construct validity

Convergent and divergent validity analyses are presented in Table 3. The findings indicate a good convergent validity of the WELCH, as there was a strong positive correlation with

**Table 2  Score distributions and missing values of WELCH items at baseline and 12-month follow-up.**

| | Baseline | | | | | | 12-month follow-up | | | | | |
|---|---|---|---|---|---|---|---|---|---|---|---|---|
| | TeGeCoach (n = 811) | | Routine care (n = 885) | | Total (n = 1,696) | | TeGeCoach (n = 588) | | Routine care (n = 645) | | Total (n = 1,233) | |
| | n | % | n | % | n | % | n | % | n | % | n | % |
| *Item 1* | | | | | | | | | | | | |
| *Impossible* | 13 | 1.6 | 4 | 0.5 | 17 | 1.0 | 16 | 2.7 | 5 | 0.8 | 21 | 1.7 |
| *30 seconds* | 9 | 1.1 | 6 | 0.7 | 15 | 0.9 | 5 | 0.9 | 11 | 1.7 | 16 | 1.3 |
| *1 minute* | 27 | 3.3 | 25 | 2.8 | 52 | 3.1 | 10 | 1.7 | 21 | 3.3 | 31 | 2.5 |
| *3 minutes* | 98 | 12.1 | 112 | 12.7 | 210 | 12.4 | 43 | 7.3 | 78 | 12.1 | 121 | 9.8 |
| *10 minutes* | 239 | 29.5 | 240 | 27.1 | 479 | 28.2 | 118 | 20.1 | 160 | 24.8 | 278 | 22.5 |
| *30 minutes* | 204 | 25.2 | 234 | 26.4 | 438 | 25.8 | 133 | 22.6 | 155 | 24.0 | 288 | 23.4 |
| *1 hour* | 139 | 17.1 | 163 | 18.4 | 302 | 17.8 | 168 | 28.6 | 134 | 20.8 | 302 | 24.5 |
| *3 h or more* | 60 | 7.4 | 79 | 8.9 | 139 | 8.2 | 73 | 12.4 | 59 | 9.1 | 132 | 10.7 |
| *Total* | 789 | 97.3 | 863 | 97.5 | 1,652 | 97.4 | 566 | 96.3 | 623 | 96.6 | 1,189 | 96.4 |
| *NAs* | 22 | 2.7 | 22 | 2.5 | 44 | 2.6 | 22 | 3.7 | 22 | 3.4 | 44 | 3.6 |
| *Item 2* | | | | | | | | | | | | |
| *Impossible* | 22 | 2.7 | 17 | 1.9 | 39 | 2.3 | 11 | 1.9 | 14 | 2.2 | 25 | 2.0 |
| *30 seconds* | 19 | 2.3 | 19 | 2.1 | 38 | 2.2 | 9 | 1.5 | 20 | 3.1 | 29 | 2.4 |
| *1 minute* | 60 | 7.4 | 77 | 8.7 | 137 | 8.1 | 32 | 5.4 | 55 | 8.5 | 87 | 7.1 |
| *3 minutes* | 199 | 24.5 | 192 | 21.7 | 391 | 23.1 | 86 | 14.6 | 129 | 20.0 | 215 | 17.4 |
| *10 minutes* | 249 | 30.7 | 262 | 29.6 | 511 | 30.1 | 153 | 26.0 | 168 | 26.0 | 321 | 26.0 |
| *30 minutes* | 134 | 16.5 | 169 | 19.1 | 303 | 17.9 | 134 | 22.8 | 128 | 19.8 | 262 | 21.2 |
| *1 hour* | 82 | 10.1 | 96 | 10.8 | 178 | 10.5 | 106 | 18.0 | 78 | 12.1 | 184 | 14.9 |
| *3 h or more* | 26 | 3.2 | 33 | 3.7 | 59 | 3.5 | 39 | 6.6 | 29 | 4.5 | 68 | 5.5 |
| *Total* | 791 | 97.5 | 865 | 97.7 | 1,656 | 97.6 | 570 | 96.6 | 621 | 96.3 | 1,191 | 96.6 |
| *NAs* | 20 | 2.5 | 20 | 2.3 | 40 | 2.4 | 18 | 3.1 | 24 | 3.7 | 42 | 3.4 |
| *Item 3* | | | | | | | | | | | | |
| *Impossible* | 104 | 12.8 | 111 | 12.5 | 215 | 12.7 | 47 | 8.0 | 90 | 14.0 | 137 | 11.1 |
| *30 seconds* | 62 | 7.6 | 71 | 8.0 | 133 | 7.8 | 33 | 5.6 | 52 | 8.1 | 85 | 6.9 |
| *1 minute* | 145 | 17.9 | 143 | 16.2 | 288 | 17.0 | 72 | 12.2 | 90 | 14.0 | 162 | 13.1 |
| *3 minutes* | 216 | 26.6 | 217 | 24.5 | 433 | 25.5 | 125 | 21.3 | 144 | 22.3 | 269 | 21.8 |
| *10 minutes* | 149 | 18.4 | 194 | 21.9 | 343 | 20.2 | 139 | 23.6 | 131 | 20.3 | 270 | 21.9 |
| *30 minutes* | 78 | 9.6 | 79 | 8.9 | 157 | 9.3 | 86 | 14.6 | 70 | 10.9 | 156 | 12.7 |
| *1 hour* | 32 | 3.9 | 29 | 3.3 | 61 | 3.6 | 60 | 10.2 | 35 | 5.4 | 95 | 7.7 |
| *3 h or more* | 5 | 0.6 | 17 | 1.9 | 22 | 1.3 | 6 | 1.0 | 7 | 1.1 | 13 | 1.1 |
| *Total* | 791 | 97.5 | 861 | 97.3 | 1,652 | 97.4 | 568 | 96.6 | 619 | 96.0 | 1,187 | 96.3 |
| *NAs* | 20 | 2.5 | 24 | 2.7 | 44 | 2.6 | 20 | 3.4 | 26 | 4.0 | 46 | 3.7 |
| *Item 4* | | | | | | | | | | | | |
| *Much slower* | 120 | 14.8 | 131 | 14.8 | 251 | 14.8 | 62 | 10.5 | 93 | 14.4 | 155 | 12.6 |
| *Moderately slow* | 320 | 39.5 | 362 | 40.9 | 682 | 40.2 | 182 | 31.0 | 245 | 38.0 | 427 | 34.6 |
| *A bit slower* | 231 | 28.5 | 250 | 28.2 | 481 | 28.4 | 186 | 31.6 | 179 | 27.8 | 365 | 29.6 |
| *At the same speed* | 107 | 13.2 | 112 | 12.7 | 219 | 12.9 | 125 | 21.3 | 90 | 14.0 | 215 | 17.4 |
| *Faster* | 23 | 2.8 | 26 | 2.9 | 49 | 2.9 | 23 | 3.9 | 17 | 2.6 | 40 | 3.2 |
| *Total* | 801 | 98.8 | 881 | 99.5 | 1,682 | 99.2 | 578 | 98.3 | 624 | 96.7 | 1,202 | 97.5 |
| *NAs* | 10 | 1.2 | 4 | 0.5 | 14 | 0.8 | 10 | 1.7 | 21 | 3.3 | 31 | 2.5 |

Rezvani et al. (2021), *PeerJ*, DOI 10.7717/peerj.12039

**Table 3  Correlation between WELCH scores and other measures of walking impairment at baseline and 12-month follow-up.**

| | | | | 12-month follow-up | | | | | | | | |
| *Correlation with WELCH score* | Baseline | | | TeGeCoach | | | Routine care | | | Total | | |
| | *n* | r | Bootstrapped 95%-CI | *n* | r | Bootstrapped 95%-CI | *n* | r | Bootstrapped 95%-CI | *n* | r | Bootstrapped 95%-CI |
|---|---|---|---|---|---|---|---|---|---|---|---|---|
| WIQ walking distance | 1564 | .65 | .62–.68 | 532 | .71 | .67–.74 | 578 | .69 | .64–.73 | 1110 | .70 | .68–.73 |
| WIQ walking speed | 1575 | .68 | .65–.71 | 531 | .73 | .69–.76 | 584 | .71 | .67–.75 | 1115 | .72 | .69–.75 |
| WIQ stair climbing | 1590 | .56 | .53–.59 | 535 | .60 | .55–.64 | 587 | .60 | .54–.64 | 1122 | .60 | .56–.63 |
| WIQ total | 1472 | .70 | .68–.73 | 493 | .75 | .72–.79 | 544 | .73 | .69–.76 | 1037 | .74 | .72–.77 |
| VascuQoL Activity | 1660 | .61 | .58–.64 | 571 | .67 | .63–.71 | 622 | .64 | .60–.68 | 1,193 | .66 | .63–.69 |
| GAD-7 | 1629 | −.14 | −.19 to −.09 | 559 | −.22 | −.28 to −.14 | 603 | −.21 | −.28 to −.13 | 1162 | −.22 | −.27 to −.16 |

the WIQ distance subscale (baseline: $n = 1,564$, $r = 0.65$, 95% CI [.62–.68], $p <$.001; 12 months: $n = 1,110$, $r = .70$, 95% CI [.68–.73], $p < .001$); a strong positive correlation with the WIQ speed subscale ($n = 1,575$, $r = .68$, 95% CI [.65–.71], $p < .001$; 12 months: $n = 1,115$, $r = .72$, 95% CI [.69–.75], $p < .001$); a strong positive correlation with the WIQ stair climbing subscale ($n = 1,590$, $r = .56$, 95% CI [.53–.59], $p < .001$; 12 months: $n = 1,122$, $r = .60$, 95% CI [.56–.63], $p < .001$); a strong positive correlation with the WIQ total score ($n = 1,472$, $r = .70$, 95% CI [.68–.73], $p < .001$; 12 months: $n = 1,037$, $r = .74$, 95% CI [.72–.77], $p < .001$); and a strong positive correlation with the VascuQoL-25 activity scale ($n = 1,660$, $r = .61$, 95% CI [.58–.64], $p < .001$; 12 months: $n = 1,193$, $r = .66$, 95% CI [.63–.69], $p < .001$). Furthermore, in absolute terms, there was a weaker correlation between the WELCH and the GAD-7 (baseline: $n = 1,629$, $r = -.14$, 95% CI [$-.19$ to $-.09$], $p < .001$; 12 months: $n = 1,162$, r $= -0.22$, 95% CI [$-.27$ to $-.16$], $p < .001$), indicating adequate divergent validity. When separated by study group at 12 months of follow-up (TeGeCoach; routine care), the associations between the WELCH and the comparator instruments were nearly identical (see Table 3), demonstrating satisfactory construct validity regardless of treatment status.

Poor to moderate QOL (VascuQoL-25 < 4; $n = 383$; $M = 14.95$; SD $= 12.20$) has been reported by 23% of the sample at baseline. A significant mean score difference of 17.3 with a very large effect size ($d = 0.96$; 95% CI [0.84–1.01]) was identified between PAD patients with high ($n = 1,277$; $M = 32.23$; SD $= 19.38$) and those with poor to moderate QOL, t $(1,001) = 20.91$, $p < .001$, indicating excellent known-groups validity for the WELCH. In multiple linear regression analysis, a logarithmic transformation was performed to correct for heteroscedasticity. Health-related QOL remained a significant predictor of the WELCH score even after controlling for potential confounders, t $(847) = 13.67$, $p < .001$, which included age, BMI, income, comorbid diseases, medication, gender, education, revascularization and heart rate training. The partial r for predicting WELCH score from health-related quality of life (pr $= .43$) was not substantially different from the zero-order Pearson's r without controlling for confounders ($r = .47$). Similar results were found at 12-month follow-up irrespective of study group (not reported).

### Responsiveness (TeGeCoach home-based exercise)

From baseline to 12-month follow-up, the WELCH [range: 0–100] improved by 6.61 points (SD $= 17.04$, 95% CI [5.12–8.10], $d = 0.39$) in the TeGeCoach group, from 28.86 (SD $=$ 18.98) to 35.47 (SD $= 22.29$). During the same period, the WIQ distance, speed and stair climbing subscale scores [range: 0–100] improved in the TeGeCoach group by 10.65 (SD $= 24.27$, 95% CI [8.44–12.86], $d = 0.44$), 6.83 (SD $= 19.55$, 95% CI [5.05–8.61], $d = 0.35$) and 5.75 points (SD $= 20.51$, 95% CI [3.90–7.60], $d = 0.28$), respectively. The WIQ total score [range: 0–100] improved by 7.66 points (SD $= 17.45$, 95% CI [5.96–9.37], $d = 0.44$), and the VascuQOL [range: 0–7] by 0.32 points (SD $= 0.81$, 95% CI [0.25–0.39], $d = 0.40$).

Figure 2 presents the ROC curves generated for the WELCH for small (+5% change), moderate (+25% change) and large MCIDs (40% change) on the WIQ, and responsiveness statistics are reported in detail in Table 4. The AUC for small changes was .66 (SE $= .02$, 95% CI [.62–.71]) for the distance subscale, .64 (SE $= .03$, 95% CI [.59–.69]) for the speed

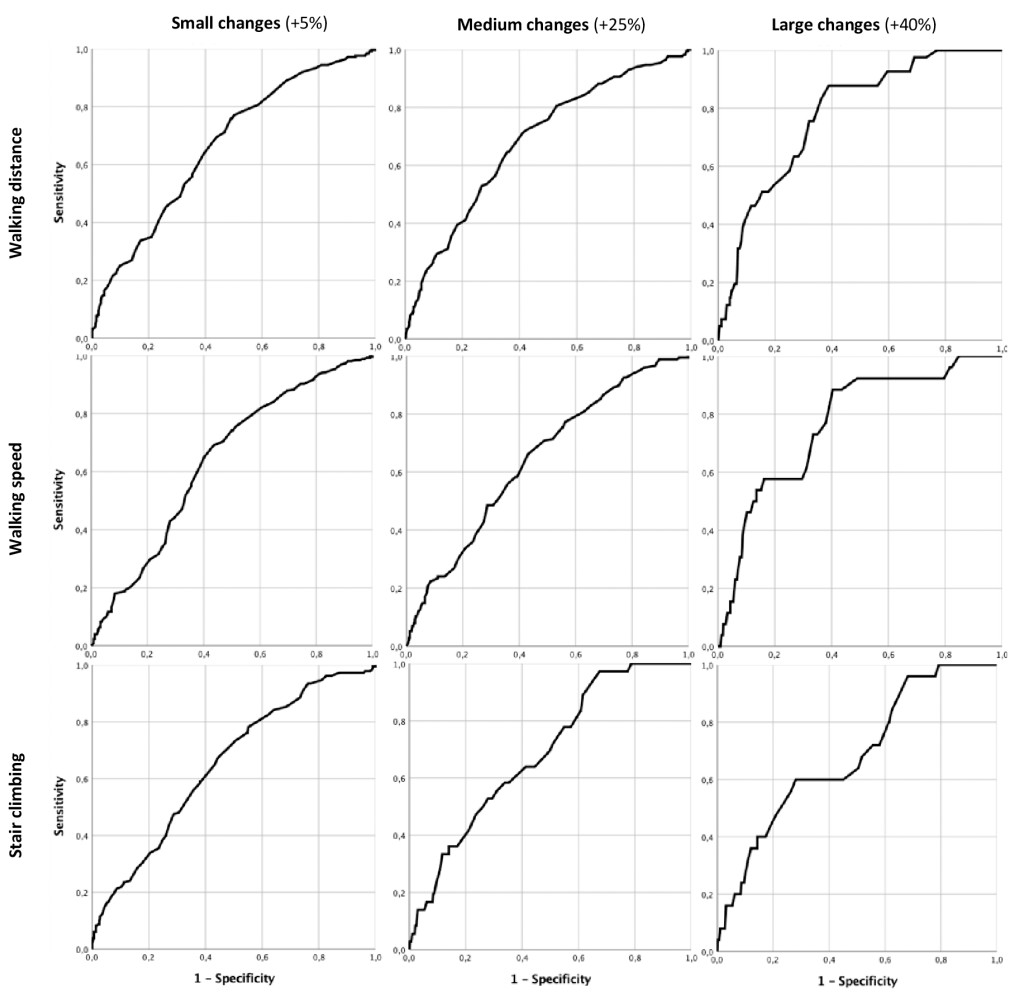

**Figure 2  WELCH ROC curves for small, moderate and large changes on the WIQ subscales.**

subscale and .65 (SE = .03, 95% CI [.60–.70]) for the stair climbing subscale. For moderate changes, the AUC was .69 (SE = .03, 95% CI [.64–.73]) for the distance subscale, .64 (SE = .03, 95% CI [.59–.69]) for the speed subscale and .69 (SE = .04, 95% CI [.61–.77]) for the stair climbing subscale. The AUC for large changes was .78 (SE = .03, 95% CI [.71–.84]) for the distance subscale, .77 (SE = .05, 95% CI [.68–.86]) for the speed subscale and .68 (SE = .05, 95% CI [.58–.79]) for the stair climbing subscale. Another ROC curve has been generated to test the ability of the WELCH to discriminate patients between study groups at 12-month follow-up (routine care *vs.* TeGeCoach). This showed an AUC of .63 (SE = .02, 95% CI [.59–.66]).

## DISCUSSION

Further psychometric validation of PROMs that measure walking capacity in PAD patients is required. This study sought to validate the German version of the WELCH, which was

**Table 4  WELCH responsiveness statistics at various thresholds reflecting small, moderate and large changes on the WIQ subscales.**

| WIQ subscale | Small (+5% change) | | | Moderate (+25% change) | | | Large (+40% change) | | |
|---|---|---|---|---|---|---|---|---|---|
| | Walking distance | Walking speed | Stair climbing | Walking distance | Walking speed | Stair climbing | Walking distance | Walking speed | Stair climbing |
| *Threshold* [a] | $\geq 5$ | $\geq 2$ | $\geq 12$ | $\geq 15$ | $\geq 11$ | $\geq 35$ | $\geq 42$ | $\geq 37$ | $\geq 41$ |
| *AUC* | 0.66 | 0.64 | 0.65 | 0.69 | 0.64 | 0.69 | 0.78 | 0.77 | 0.68 |
| *SE* | 0.02 | 0.03 | 0.03 | 0.03 | 0.03 | 0.04 | 0.03 | 0.05 | 0.05 |
| *95% CI* | 0.62–0.71 | 0.59–0.69 | 0.60–0.70 | 0.64–0.73 | 0.59–0.69 | 0.61–0.77 | 0.71–0.84 | 0.68–0.86 | 0.58–0.79 |
| *n positive* | 249 | 256 | 183 | 170 | 175 | 36 | 41 | 26 | 25 |
| *n negative* | 257 | 250 | 323 | 336 | 331 | 470 | 465 | 480 | 481 |

**Notes.**

[a]Adopted from *Gardner, Montgomery & Wang (2018)*.

developed to address the limitations of existing PROMs for the measurement of walking impairment in PAD patients.

Consistent with previous findings (*Ouedraogo et al., 2013*), few missing values and > 95% completely filled out questionnaires largely support the excellent feasibility of the WELCH in PAD patients. The brief nature of the questionnaire with only four items, and that it can be easily completed without external support, makes it particularly attractive compared to other questionnaires, especially in settings where time pressure is high (*e.g.*, doctor's office). Furthermore, there were no floor or ceiling effects, which enables the WELCH to discriminate equally between symptomatic PAD patients across the entire IC severity spectrum.

In agreement with previous studies (*Cucato et al., 2016*; *Ouedraogo et al., 2011*), the WELCH has provided evidence for good psychometric properties in terms of test-retest reliability. This finding is directly related to the usefulness of the WELCH in repeated measurement designs ensuring that scores changes are due to real changes rather than irrelevant artefacts, making an important contribution to its psychometric validity and reliability.

In terms of construct validity, results were in line with previous studies (*Ouedraogo et al., 2013*; *Abraham et al., 2014b*; *Tew et al., 2014*), providing further evidence that the WELCH has good psychometric properties that make it suitable for use in assessing walking impairment living with intermittent claudication. As would be expected, the WELCH demonstrated satisfactory convergent validity, revealing a consistent pattern of moderate to strong correlations with the criterion measures. The associations between the WELCH and others PROMs indicate that the WELCH reflects similar but not identical constructs of walking impairment, with the WELCH showing the highest agreement with measures reflecting walking distance and walking speed (WIQ subscales). Notably, the correlation coefficients between the WELCH score and the WIQ subscales are fairly similar to those observed between the WELCH and treadmill maximum walking distance in previous studies (*Ouedraogo et al., 2013*; *Abraham et al., 2014b*; *Tew et al., 2014*), which further supports the validity of the WELCH for assessing walking impairment in symptomatic PAD patients. Simultaneously, the WELCH also shows a high correlation with health-related physical

functioning (VascuQoL-25 activity subscale), indicating that the WELCH also quantifies the subjective patient experience by reflecting walking limitations in daily life. Furthermore, the WELCH was able to very accurately discriminate between patients with poor and high levels of health-related QOL, demonstrating excellent known-groups validity. Although the relationship between QOL and walking impairment is already well known (*Golledge et al., 2020*), these results indicate that the WELCH also indirectly reflects aspects of QOL in symptomatic PAD patients, further supporting the WELCH's value as a patient-relevant outcome measure by addressing the impact of PAD on those living with the disease. Finally, as predicted from previous findings (*Smolderen et al., 2009*), the WELCH demonstrated good divergent validity in relation to anxiety symptoms (GAD-7), likewise supporting the validity of the instrument.

To date, only few PROMs for PAD patients have been studied in terms their responsiveness, which is a major shortcoming since PROMs are frequently used for measuring the effect of treatment in research and clinical practice (*Conijn et al., 2015*), raising doubts on the validity of results. Likewise, evidence on the responsiveness of the WELCH is sparse (*Henni et al., 2019*). In agreement with the construct validity results, the WELCH was found capable to detect large clinically important improvements observed in walking distance and speed following a home-based training regimen, suggesting that the WELCH may be considered responsive to exercise interventions, whereas small to moderate improvements did not generate sufficient change on the WELCH. These findings have direct implications for its use in therapy settings, as they show that the threshold for detecting clinically important effects is relatively high when using the WELCH. The ability to detect a restricted range of clinically meaningful changes (+40%) in response to exercise interventions may limit its utility in clinical settings, particularly since exercise interventions generally have small to moderate, yet clinically meaningful effects on walking impairment (*Golledge et al., 2019*; *Parmenter, Dieberg & Smart, 2015*). The WELCH may therefore be better suited to capture improvements after combined therapies of IC (*i.e.*, exercise therapy plus lower extremity revascularization), as greater improvements in walking performance are usually achieved than after either therapy alone (*Biswas et al., 2021*). The limited responsiveness found here is also comparable to the responsiveness of the WELCH after performing revascularization, where a moderate correlation with treadmill maximum walking distance was shown (*Henni et al., 2019*). In addition, the 12-month time interval between the measurements may have also reduced the WELCH's ability to detect improvements after exercise therapy, as the WELCHs responsiveness tends to decrease over time (*Henni et al., 2019*). Despite the promising results reported, the responsiveness of the WELCH in relation to other walking impairment measures (*i.e.*, other PROMs, functional testing), and whether it depends on the time interval between measurements (*i.e.*, short-term, long-term change), mode of intervention (*i.e.*, invasive, non-invasive) and degree of change (*i.e.*, small, medium, large intervention effects) still remains to be conclusively determined in further studies.

The present study is the first study to evaluate the psychometric properties of the WELCH in a German clinical population and has several strengths compared to previous validation studies. With a patient to item ratio of 400:1, this is the largest validation study

of the WELCH. Furthermore, this is the first study to assess the psychometric properties of a PROM for assessing IC based on the COSMIN checklist, which established guidelines for the psychometric validation of health status PROMs (*Terwee et al., 2007*). Although assessing psychometric properties of health status PROMs is common practice, the study quality in the field of PAD is often inadequate (*Conijn et al., 2015*), which underlines the importance of adopting universal quality criteria. The COSMIN checklist provided a rigorous methodological structure that helped in minimizing methodological bias. It would therefore be useful to use the COSMIN checklist to further evaluate the measurement properties of PROMs for PAD patients.

Several limitations of the study should be noted, including that the translation process did not rigorously comply with the Principles of Good Practice for the Translation and Cultural Adaptation Process for PROMs (*Wild et al., 2005*). Despite this shortcoming, the present study confirms the psychometric validity of the German version of the WELCH, suggesting that the translation and cultural adaptation process can be considered acceptable.

Furthermore, the comparator instruments used to test construct validity are not gold standards, which may have reduced the observed correlations. Notwithstanding this, the WELCH showed high correlations with the comparator instruments, as expected, with the strongest associations on the distance and speed subscales of the WIQ, supporting the validity of the WIQ in assessing walking impairment (*McDermott et al., 1998*; *Sagar et al., 2012*; *Tew et al., 2013*; *Nicolai et al., 2009*) and thus being well suited as a valid comparator instrument. To provide further evidence for the construct validity of the WELCH, it should also be tested against third-party assessments (*e.g.*, ratings by health professionals) and gold standard measurements (*e.g.*, treadmill testing) in future validation studies.

## CONCLUSIONS

This article provides evidence that the German version of the WELCH questionnaire is a valid instrument for assessing walking impairment in patients with intermittent claudication. The WELCH, when used appropriately, enables the assessment of walking impairment in PAD patients, while compensating for the existing limitations of existing PROMs. In view of the excellent feasibility and good construct validity, its use can be recommended in clinical settings, as the medical team can quickly gain insight into the PAD patient's walking impairment condition without the need for cumbersome and time-consuming functional assessments. Nonetheless, despite its practicality, the WELCH should be treated with a degree of caution when used to evaluate the benefits of exercise treatments in clinical trials and practice. Alternatively, the WELCH merits consideration in vascular surgery to measure changes evoked by combined treatments (*i.e.,* exercise therapy plus lower extremity revascularization), which, however, remains to be investigated in future studies.

## ACKNOWLEDGEMENTS

We would like to thank the authors of the WELCH questionnaire, who kindly provided us with the German version of the questionnaire. We would also like to thank Finja

Mäueler, who assisted us with great enthusiasm and commitment in the preparation of the manuscript.

### Funding

This study, which was conducted as part of a clinical trial (trial registration: NCT03496948 on CT.gov), is funded by the German Innovation Fund (01NVF17013) of the Federal Joint Committee (G-BA), the highest decision-making body of the joint self-government of physicians, dentists, hospitals and health insurance funds in Germany. The use of grant funds is monitored by the German Aerospace Center (DLR). The funders had no role in study design, data collection and analysis, decision to publish, or preparation of the manuscript.

### Grant Disclosures

The following grant information was disclosed by the authors:
Federal Joint Committee (G-BA): 01NVF17013.

### Competing Interests

The authors declare there are no competing interests.

### Author Contributions

- Farhad Rezvani conceived and designed the experiments, performed the experiments, analyzed the data, prepared figures and/or tables, authored or reviewed drafts of the paper, and approved the final draft.
- Martin Härter and Jörg Dirmaier conceived and designed the experiments, authored or reviewed drafts of the paper, and approved the final draft.

### Human Ethics

The following information was supplied relating to ethical approvals (i.e., approving body and any reference numbers):

The Medical Association of Hamburg granted Ethical approval to carry out the study (Ethical Application Ref: PV5708).

### Data Availability

The measurements for the first (t0) and second (t1) time points are available in the Supplementary File.

### Supplemental Information

Supplemental information for this article can be found online at http://dx.doi.org/10.7717/peerj.12039#supplemental-information.

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
