# Peer review of "Measuring walking impairment in patients with intermittent claudication: psychometric properties of the Walking Estimated-Limitation Calculated by History (WELCH) questionnaire"

_PeerJ, doi:10.7717/peerj.12039_

## Round 0.1 · original submission · Major Revisions

Two reviewers have commented on your manuscript and made some points that will need to be addressed. Our collective impressions (theirs and mine) suggests that this work has the potential to be useful to researchers in this area and so I will invite you to prepare a response to each of their comments, and mine, with revisions tracked in response to each point (accompanied by a brief explanation of the revision(s)) or an explanation of why no changes have been made for that point.

Reviewer #1 notes that the dataset could be more accessible to readers. Readers should be able to use this data, along with the text in your statistical methods, to reproduce the values in your manuscript. While a semi-colon delimited file is fine, I would suggest another extension than .csv (comma-separated) in that case. This needs to be accompanied by a clear codebook explaining the meaning of each variable (this would perhaps be easier with English variable names in the data file) and the levels of the categorical variables. An Excel workbook has the advantage of allowing you to package this as a separate worksheet alongside the data, but an SPSS .sav file would perhaps be easiest to combine these two things and so make it more likely that your results can all be reproduced. I also agree with the Reviewer that any discrepancies between the trial registration, protocol article, and the present manuscript need to be identified and explained. Changes between these are not inherently problematic, but it is vital that readers are aware of these changes and the reasons for them, so that they can judge whether this might affect the study’s (internal and external) validity. They request some additional information about methods that I think readers will appreciate you providing, including how you determined your interpretation of the psychometric statistics (with references for the reader’s benefit). A flowchart might be a useful way of presenting the study participants and reconciling the various sample sizes they note, as well as identifying the reasons for loss to follow-up, which some readers here might be interested in. I think that their other comments are also worthy of careful consideration and response.

Reviewer #2 makes an important point about breaking up Table 1 into subgroups. I’ll expand on their point here below as I think it could be useful to readers if you describe those lost to follow-up and in the intervention group. Their point about the effect of measurement error on the correlation coefficients is well made (with the correlations being attenuated in the presence of measurement error) and this might be worth further comment in the discussion if you agree.

As some additional comments, I would caution against absolute statements (e.g. “no floor or ceiling effects” on Lines 26 and 184–185, “which however has not yet been psychometrically validated” on Line 81, and “Test-retest reliability has not yet been tested for the WELCH” on Lines 189–190) as (as demonstrated by Reviewer #1) it takes only a single counterexample to disprove such statements. I suggest qualifying these with “…had no strong…”, “To the best of our knowledge” or “As far as we are aware”, and similar and as appropriate, so that the truth of the statements is not threatened by other research. A similar point applies to “being the first study” (Lines 226–227), but here I would not consider this as a strength (where strengths are for me reasons to believe the results of the study, i.e. supporting the internal and/or external validity of the findings). Novelty is certainly a motivation for performing a study, but does not in itself provide greater confidence in the conclusions and I suggest separating this point, which is appropriate, from the strengths.

I would like to see more CIs in the abstract, as you do on Line 27 but not on Lines 30, 31, 32, 33, 34, and 35. I appreciate that where you give an interval of correlations this might preclude giving the 95% CIs, but where these can be added to the abstract, the reader will benefit in their understanding of your findings. The same point applies to the results in the body of the manuscript, where you provide the 95% CI on Line 152 but not for the subsequent results until Line 165 and not after this line as far as I can see. The same applies to Table 3.

I would be careful about describing items as using a “Likert scale” (e.g. Line 84). This overloads the word “scale” (which also refers to a collection of items measuring the same construct) and “Likert”, which claims equi-spaced options (providing interval-level measurement properties) and is difficult to establish and seldom done in practice. See http://www.john-uebersax.com/stat/likert.htm for a discussion of these and other issues. I suggest referring to the items as “eight-point ordinal items”, etc. unless the Likert aspect has been established (your description on Line 107 also avoids this issue). I’d suggest giving a very brief explanation of how missing values were handled in your manuscript (Lines 86–87) as a kindness to readers.

I’d like to see you justify your choice of ICC (Lines 121–122) as the “usual” choice here in my view would be the two-way mixed effects model (see DOI 10.1016/j.jcm.2016.02.012, particularly Figure 1).

I would also like to see a (brief) description of the statistical model checks/diagnostics. For example, you are assuming linearity with the correlations, normally distributed errors for the t-tests, etc. How were these assessed? Note that I am assuming your level of significance on Line 139 is two-sided, but the sidedness of tests should be made explicit to avoid readers needing to make such assumptions.

While comparing those with high and low HRQoL provides supportive evidence of validity here, I wondered if you’d considered adjusting for known/potential confounders, e.g. age and sex, to reduce the risk of confounding in this analysis? Was information on ethnicity, which could be another potential confounder, collected?

Finally, for the tables, following up on Reviewer #2’s comment, I’d like to see a comparison of those at baseline only and those followed through to 12 months. Table 1 could show both groups here, allowing the reader to consider the missing data mechanisms. Note that “range” is strictly speaking the difference between the smallest and largest values (see https://en.wikipedia.org/wiki/Range_(statistics)) and I’d label these as minimum–maximum instead. Also, are you sure that it was gender (the social construct) that was measured and not sex (the biological state)? There didn’t seem to be any explanation of how this data was obtained/this question was asked and this could be added to the methods (my apologies if I’m overlooking something here). I think two decimal places for the mean and SD of both age and BMI (and the minimum and maximum BMI) might be excessive. Table 2 could similarly present the baseline data for the n=1279 at 12 months. When the reader sees Table 3, they might wish to see the demographics in Table 1 further broken down for this group, but you might feel that this would make Table 1 unwieldy. The transitions between the levels of responses at baseline and 12 months should be shown visually (think of either a spaghetti plot for the summary score or transition (Markov chain) diagrams for the individual responses for each item) to further explain the responsiveness beyond that shown by looking at the mean and standard deviation of changes. This last point is merely a suggestion for you to consider.

·

Basic reporting

1) A dataset has been submitted with the article, however it is largely incomprehensible. The file should be edited so that it is much easier to interpret/navigate/use.
2) It would be useful to indicate somewhere in the abstract that the sample consisted of intermittent claudication patients specifically.
3) Line 51: define PROM in full at first use
4) Table 3 title: “exercise intervention group (n = 994)” – is this n-value correct? It seems too large and not in keeping with the n-values included in the table itself.
5) P177: “inevitable” – seems an odd choice of word since I don’t think it is inevitable. How about replacing with “required” or “necessary”?
6) References 7 and 12 are the same.

Experimental design

1) Line 65: The WELCH is listed as a secondary outcome measure on the clinicaltrials.gov registration but not in the BMJOpen protocol paper; why was it not included in the latter?
2) Line 68: The sample of 2042 doesn’t match with the sample size mentioned on the registration page (1982) or protocol paper (1760). Please explain the discrepancy. I presume that it’s something to with needing to enrol 2042 to end up with enough people also completing baseline (1759).
3) Line 73-74: Please can you explain the data collection people in more detail as this will aid the interpretation of feasibility (e.g. rate of missing data). You say the participants received a battery of paper questionnaires. Was this handed to them in-person or mailed out to them? They return by mail but was a pre-paid envelope used for this? What support, if any, could the participants access with completing the questionnaires? What processes did the trial use for minimising missing data? E.g. were reminders sent to people who didn’t respond, could an investigator chase missing data via telephone?
4) Line 80-81: The original authors of the WELCH are French, so I’m wondering what the translation process was when creating the German version, e.g. who was involved and how rigorous was the process to ensure appropriate wording? This is referred to in the discussion but should probably also be clarified in the methods. The discussion indicates that the questionnaire was translated and back-translated – can the authors verify this as it would help address the first limitation in the discussion? Please also state in the text that the German version used can be viewed as a supplementary file.
5) Line 110-111: The COSMIN standards set does not appear as a supplementary file as indicated on these lines.
6) Line 114-117, 123, 127, 132, 136: Where do the thresholds used for assessing feasibility, floor and ceiling effects, test-retest reliability, validity, and responsiveness come from? Are they completely arbitrary or can you justify and add supporting references? Please also clarify if these thresholds were specified a priori or after the data had been observerd/inspected.
7) Line 118-119: For test-retest reliability, 72 PAD patients completed the WELCH twice within a 2-week period around the 12-month follow-up point in the RCT. Did your ethics approval cover the repeat assessment here? There’s no mention of it in the trial registration or protocol paper.

Validity of the findings

1) Table 1: Can you provide any data on PAD status, e.g. ABI or proportion of patients in Fontaine IIa and IIb?
2) Line 189-190: “Test-retest reliability has not yet been tested for the WELCH” – this is not true. It was definitely tested in Ouedraogo et al. (2013) and Cucato et al. (2016). Please revise this section of the discussion accordingly.
3) Results: please add summary statistics for the total WELCH score at baseline and follow-up. You only show the mean (SD) change score.
4) Line 215-216: “… it is likely that changes in walking distance are not well captured by the WELCH” – I’m not sure I agree with this and it seems like speculation. It is OK to speculate, but it should be identified as such. Inspection of table 2 suggests that it’s a combination of changes across all four items that would contribute to a change in the total score. Items 1-3 ask about the continuous duration achievable at different walking speeds. Wouldn’t an increased ability to walk for a longer duration reflect an increased ability to walk a greater distance?

·

Basic reporting

Rezvani and colleagues perform a psychometric analysis of the German version of the « walking estimated calculated by history (WELCH) questionnaire in a rather large. The study aimed at analyzing various (feasibility, test-retest reliability, construct validity (i.e. convergent, divergent and known-group validity), and sensitivity to changes properties in comparison to other existing tools (i.e.: VascuQoL, GAD-7 and WIQ). Although not detailed in the method section, according to the data file provided for review, patients in the follow up group seem to results from two different populations including both patients completing the questionnaire twice without apparent therapeutic intervention (Routineversorgung : routine treatment) and patients that underwent an intensive rehabilitation program teGeCoach that seems an active telemonitored intermittent walking exercise.
The authors should be congratulated for the quality of the writing and organization of the paper. The goal of the study is original and adds to the existing literature.

Experimental design

The goal of the study is original and adds to the existing literature and seems to me in the general scope of the journal for medical and health sciences.

Validity of the findings

I would strongly recommend that the two populations should be described in a new table after table 1. I completely agree with the authors that the questionnaires may be analyzed for the whole group of 1759 included patients at baseline and/or 1279 at twelve months, when the aim is to analyze feasibility, test-retest reliability, construct validity. Nevertheless, I do believe that analyzing separately these two subgroups would provide some additional useful information to the paper.

Indeed, in studying changes in scores, the variability in score differences (M12-M0) has mathematically variability for each toll that is twice the variability of the toll against a control gold standard (as could be a 6 min walking tests or treadmill). Thereby the coefficient of correlation of the linear regression analysis of the score differences is as expected quite low because of the high residuals that would be expected for each tool against a standard. It seems to me that the conclusion that the authors made that the WELCH is relatively insensitive to changes as compare to the WIQ could be strengthened by comparing the area under ROC curve of each of the two tolls in determining if patients where actively treated or not. Indeed, lacking a gold standard, the toll that have the higher area is expected to be the best one for estimating changes in case of active treatment.

Additional comments

No further comment

---

## Round 0.2 · Minor Revisions

Thank you for your revised manuscript and responses. You’ll see that both reviewers are happy with the new version of your manuscript, with just a small stylistic comment from Reviewer #2. Well done!

I agree with you that the new Table 1 is unwieldy (despite being informative) and I think your solution to the Table 1 problem would be the best one here (the “short version” being included in the manuscript itself and the “long version” in the supplementary materials).

If you can address the Table 1 point and a few minor comments from myself (listed below), I will be delighted to accept your manuscript.

Line 98: “participants who have” => “participants who had” (to match “received” and “were”).

Line 100: “questionnaires have been” => “questionnaires had been” (same).

Line 112: Do you mean “can maintain WALKING at different speeds”?

Line 120: I appreciate that you explain the amount of missing data in detail on Lines 216–218 and in Table 2, but I wonder if readers would appreciate a simple “< 5% missing data per item” here (as you did on Lines 31–32) to match Lines 130 and 140 just below.

Lines 159–162: I don’t know that this is exactly what Schafer is saying. They do say that “When the rate of missing information is small (say, less than 5%) then single-imputation inferences for a scalar estimand may be fairly accurate.” but this is a distinct point from considering missing data to be MAR (which is a mathematical assumption, which basically states that, on average, there is no difference between the unknown missing data and a best guess of that data based on what is known). If the point you want to make is that multiple imputation was not considered necessary, or that single imputation was considered sufficient, as long as missing data was < 5%, that’s fine, that’s just not quite how I’m reading your current text.

Line 166: You could also note that two weeks was assumed (I’m assuming!) to be long enough that they would forget their exact responses, i.e. they would not answer the second time based on memory from two weeks earlier. On Line 164, you say “within two weeks” and I wonder if giving the mean (or median) time between assessments would be useful, or the minimum and maximum if that’s more informative. You could give this information instead (or as well) on Line 225 if you preferred.

Lines 205–206: Perhaps “to allow TRACKING the pattern of missing data”? (or “to allow US to track the pattern of missing data”?)

Lines 212–214: Can you provide a reference here as re-reading this, it comes across as quite subjective?

Lines 239–240: Apologies for the pedantry, but you could say “there was a weaker (IN ABSOLUTE TERMS) correlation between” here to account for differences in signs or something like “there WERE STATISTICALLY SIGNIFICANT NEGATIVE correlationS between” to emphasise the different direction.

Line 253: I think that this is fine for the underlying (implicit) causal model as it seems to me that BMI is more likely to be a confounder than a collider or a mediator, but it is perhaps worth you checking that you don’t want to anticipate this point on the part of some readers and so preempt it here. This is just a suggestion for you to consider. The same might also apply to income, comorbidities, and medication. You could, if you wanted, set this up back around Line 184 instead. Again, this is just an observation and not a request for changes unless you wish.

Line 266: I’m assuming the “SE =” here should be “SD =”?

Lines 344–346: Previously I’d said “Novelty is certainly a motivation for performing a study, but does not in itself provide greater confidence in the conclusions and I suggest separating this point, which is appropriate, from the strengths.” and your response didn’t indicate that you disagree, so I’m wondering if an edit here was perhaps missed? Since novelty doesn’t affect validity, so isn’t a strength in my personal view, I wonder if “The present study is the first study to evaluate the psychometric properties of the WELCH in a German clinical population and has several strengths compared to previous validation studies.” would be an acceptable rearrangement?

Line 360: Perhaps “are noT gold standards”?

Line 362: There are two “as expected”s on this line and only one seems necessary.

Figure 1: This reads as if all 63,000 were approached (as does Line 85, giving a recruitment rate of 3.2%). Is this the case or was there an additional step (“Invited to participate (n=XXXX)”) in between?

Table 3: This is more of a typesetting question, but I wonder if the values should be right justified so that the negative sign doesn’t affect the alignment of the decimal points.

·

Basic reporting

no comment

Experimental design

no comment

Validity of the findings

no comment

Additional comments

I think that the authors have responded well to the comments raised in the initial review. The manuscript appears to have been strengthened and will be a useful addition to the evidence base.

·

Basic reporting

The authors have performed an extensive revision of the initial manuscript accounting for the suggestions atht were made by the editor and the two reviewers and improved their manuscript significantly

Experimental design

The study and experimental design is clear and well defined and fills the technical and ethical standard.

Validity of the findings

I have no further comments

Additional comments

Thank you for the careful correction of your manuscript and for the interesting results that you show. I look forward for future evaluation such as long term follow up...
Minor point :
Just a a doubt at line 246 that the sentence may start with a number (this might be changed at editing)

---

## Round 0.3 · accepted · Accept

Thank you for your responses and revised manuscript, which I am delighted to accept. I have noted a few, mostly very minor typographical, points, listed below, which I’ll leave up to you to address/check in the copyediting process. Please do check carefully the one about mediators on Line 186 to make sure that this is what you intended to say. Well done on what I think is a well-written and useful contribution to the literature.

Overall, the use of leading zeros seems inconsistent to me, but perhaps I’m not picking up on the rule being used. For example, the abstract contains “0.71 – 0.88” (Line 31) and “.68-.86” (Line 40). This should be standardised across the entire manuscript.

Line 21: You have a newline after the abstract heading (c.f. Lines 17, 27, and 43).

Line 37: There seems to be a spurious space before the decimal point in “- .22”.

Line 40: The CI here uses a hyphen rather than an en-dash (c.f. Lines 31, etc.)

Line 100: There are no spaces here around the en-dash in “2–4”, but generally, e.g. Line 31, you have used spaces there. Personally, I would avoid spaces around en- and em-dashes, but this should be consistent either way.

Line 112: Again, a hyphen is used to indicate a range (“1-3”) where an en-dash has generally been used. See also Lines 116, 121, 265–283, and perhaps elsewhere.

Line 186: Are you sure you mean “and mediators” (newly added to this version) here? Mediators are on the causal pathway and generally we want to make sure we don’t adjust for these. Adjusting for a mediator makes the question about the direct effect (the effect not mediated) rather than the total effect (including mediated effects). Most of the time we want to include confounders (to reduce distortion of the association of interest) and competing exposures (to reduce unexplained variability in the data) and exclude mediators (unless we are interested in the direct effect only) and colliders (as these bias associations).

Line 212: Another possible spurious space, here before the comma: “follow-up ,”

Lines 231–232: The en-dash (I think) in “test – retest reliability” should be a hyphen. C.f. Line 22, for example.